# Incidence and Trend of Type I and II Endometrial Cancer in Women from Two Population-Based European Cancer Registries (1998–2012)

**DOI:** 10.3390/ijerph19073789

**Published:** 2022-03-23

**Authors:** Daniel Ángel Rodríguez-Palacios, Sandra M. Colorado-Yohar, Michel Velten, Ricardo J. Vaamonde-Martín, Mónica Ballesta, María-Dolores Chirlaque

**Affiliations:** 1Preventive Medicine Unit, Virgen del Castillo Hospital, Avenida Feria, S/N, 30510 Murcia, Spain; daniel.rodriguez2@carm.es; 2Department of Epidemiology, Murcia Regional Health Council, Ronda de Levante, 11, 30008 Murcia, Spain; ricardoj.vaamonde@carm.es (R.J.V.-M.); monica.ballesta@carm.es (M.B.); mdolores.chirlaque@carm.es (M.-D.C.); 3CIBER Epidemiology and Public Health (CIBERESP), Instituto de Salud Carlos III (ISCIII), Avenida Monforte de Lemos 3-5, 28029 Madrid, Spain; 4Research Group on Demography and Health, National Faculty of Public Health, University of Antioquia, Calle 62 No. 52–59, Medellín 050010, Colombia; 5Biomedical Research Institute (IMIB-Arrixaca), El Palmar, 30120 Murcia, Spain; 6Department of Epidemiology and Public Health, Bas-Rhin Cancer Registry, Inserm UMR-S1113, ICANS, University of Strasbourg, 67085 Strasbourg, France; michel.velten@unistra.fr; 7Department of Health and Social Sciences, University of Murcia, Campus de Espinardo, 30100 Murcia, Spain

**Keywords:** endometrial cancer, incidence, trends, Europe

## Abstract

Endometrial cancer (EC) is the most frequent female genital tract cancer in Europe. This cohort study aimed to determine age-standardised incidence rates and long-term trends of type I and II endometrial cancer in women from population-based cancer registries in the Region of Murcia (Spain) and the Bas-Rhin area (France). Data of new cases of endometrial cancer between 1998 and 2012 were obtained from the Murcia and Bas-Rhin cancer registries. In that period, 3756 cases of endometrial cancer were recorded, with 3270 corresponding to type I EC and 486 corresponding to type II EC. The Bas-Rhin area presented higher age-adjusted incidence rates than those in the Region of Murcia for both type I EC (24.2 and 19.3 cases/100,000 person-years (py), respectively) and type II EC (4.4 and 2.3 cases/100,000 py, respectively). Joinpoint regression showed no changes in trends. In both populations, there was an increasing trend for both EC types, but the trend was steeper in the Region of Murcia and larger overall for type II EC. Finally, a significant increase was observed in the annual trend of type II EC. Further studies are warranted to determine the potential risk factors, and continued efforts are needed to improve the recording and monitoring of EC types.

## 1. Introduction

Endometrial cancer (EC) is the second most frequent gynaecological cancer in the world and the first in continental Europe, with an estimated world standardised incidence rate of 15.8 cases/100,000 women per year in 2018; in Spain and France, the incidence rates were 13.7 and 14.7 cases/100,000 person-years (py), respectively [1]. Endometrial cancer incidence has shown an increasing trend in Europe for the last three decades. However, in countries such as Spain, there was an average annual percentage change (AAPC) of 0.4% between 1994 and 2002; in other countries, such as France, a different behaviour was observed, with a continuous decline in recent years (1994–2005: AAPC −1.1%) [2,3].

Endometrial carcinoma is a hormone-dependent cancer, the aetiology of which is related to oestrogen exposure, both exogenously and endogenously [4,5,6]. In 1983, Bokhman proposed for the first time the classification of oestrogen-dependent EC (which encompasses 70–80% of carcinoma cases) as type I tumours, and oestrogen-independent EC (which corresponds to 10–20% of cases) as type II tumours. In addition, Bokhman described an unequal distribution by age of these two types of carcinomas, with type II EC being the most frequent in older women. 

The association between hormonal exposure and type I endometrial carcinoma is well known, but this is not the case for type II EC, which is less common and thus has been the subject of fewer epidemiological studies [7]. Hormonal factors associated with EC are mainly related to lifestyle, such as obesity, sedentary behaviour, the use of oestrogen-only hormone replacement therapy and the use of oral contraceptives [8,9,10,11]. It is possible that the differences in incidence are related to unequal exposure to these factors in different regions of the world.

The majority of endometrial cancers receive surgical treatment only, currently by minimally invasive surgery (which sometimes includes the robotic technique). In a number of situations where the disease has expanded, adjuvant chemotherapy and radiotherapy are also indicated. However, survival has not improved in all cases. To this end, sentinel node lymph evaluation is one of the main options under study in the last decade, replacing generalised lymphadenectomy, as it offers the possibility of selecting patients who would benefit from chemotherapy treatment, assuming a less aggressive test [12,13,14].

To determine the trends in endometrial carcinoma in two different populations in Europe, this study analysed specific cases of endometrial carcinoma, types I and II, in geographically and culturally distinct areas of southeastern Spain (Region of Murcia) and northeastern France (Département of Bas-Rhin); the data were obtained from two population-based cancer registries, with high coverage (near 100%). The main objective of this study was to determine the age-standardised incidence and the long-term (15-year period) incidence trends of EC in both populations as well as to compare the incidence in the two regions by type of endometrial carcinoma and age group.

## 2. Materials and Methods

### 2.1. Data Collection

This cohort study included data from new cases of endometrial cancer recorded between 1998 and 2012 obtained from the Murcia cancer registry (Registro de Cáncer de Murcia—RCM, Murcia, Spain) and the Bas-Rhin cancer registry (Registre des Cancers du Bas-Rhin—RCBR, Strasbourg, France), which register all cancer cases, except non-melanoma skin cancers, diagnosed in their respective areas. They are affiliated with the European Network of Cancer Registries (ENCR), so that data from these registries have been included in successive editions of ‘Cancer Incidence in Five Continents’, the reference publication edited by the International Agency for Research on Cancer (IARC, Lyon, France) (ci5.iarc.fr). The process of registering new cases of cancer occurs through notifications and active searches in public and private hospital and non-hospital health centres [15,16]. The registries collect data on the histology of each cancer, as determined by the pathologist who issued the histological report at the time of cancer diagnosis. Once cancer cases have been reviewed by qualified personnel, to verify and finalise the sociodemographic data and tumour characteristics, each case is assigned a topographic code and a morphological code based on the International Classification of Diseases for Oncology (ICD-O) in its successive editions. Registration and validation were conducted in accordance with the international recommendations of the IARC and the ENCR.

The cancer registry of the Region of Murcia, active since 1982, is affiliated with the Epidemiology Service of the Ministry of Health and included in the Spanish Cancer Registry Network (Red Española de Registros de Cáncer—REDECAN) [17]. Demographic data were obtained from the National Institute of Statistics and the Regional Statistics Centre of Murcia [18,19]. The Bas-Rhin cancer registry has been in operation since 1975 and is affiliated with the University of Strasbourg (France). In addition, it is included in the French network of cancer registries (FRANCIM) and is maintained in collaboration with the French National Public Health Agency (Santé Publique France) and the French National Cancer Institute (Institut National du Cancer—INCa) [20]. The demographic data correspond to the data provided by the National Institute of Statistics and Economic Studies (Institut National de la Statistique et des Études Économiques—INSEE) [21].

The study variables were age at diagnosis, year of diagnosis, and topographic and histological code. The latter were defined by the International Classification of Diseases for Oncology, Third Edition (ICD-O3) [22]: topographic code—malignant neoplasm of corpus uteri (C54); and its sub-locations: isthmus (C540); endometrium (C541); myometrium (C542); fundus (C543); overlapping sites (C548) and unspecified corpus uteri (C549); and histological code—malignant neoplasia (8000–8001); epithelial neoplasms (8010–8045); epidermoid neoplasms (8050–8082); adenomas and adenocarcinomas (8140–8384); mucoepidermoid neoplasms (8430–8490); complex epithelial neoplasms (8560–8580); soft tissue tumours and sarcomas (8800–8804); fibromatous neoplasms (8810–8833); myomatous neoplasms (8890–8920) and complex mixed and stromal neoplasms (8930–8991).

### 2.2. Statistical Analysis

The numbers of type I and II EC cases were analysed. For estimation of crude incidence rates, the denominator was an estimate of the resident population at mid-year, based on official figures as of 1 January. Age-adjusted incidence rates, together with their 95% confidence intervals, were calculated using the direct method (Breslow and Day, 1987), standardised to the 2013 European Standard Population [23]. In the analysis of the incidence of EC, newly recorded cases between 1998 and 2012 were considered. Age was categorised into 7 groups (≤54, 55–64, 65–74, 75–84, ≥85, <65 and ≥65 years), and year of diagnosis was grouped into 3 periods (1998–2002, 2003–2007 and 2008–2012). The histological codes were grouped into type I and type II EC based on the classification described by T. Evans et al., as it included the majority of histological codes. Type I endometrial carcinoma included adenocarcinoma (8140), solid carcinoma, NOS (8230), endometrioid carcinoma, NOS (8380), secretory endometrioid adenocarcinoma (8382), ciliated endometrioid adenocarcinoma (8383), mucinous adenocarcinoma (8480) and mucin-producing adenocarcinoma (8481). Type II EC included undifferentiated carcinoma, NOS (8020), anaplastic carcinoma, NOS (8021), papillary carcinoma, NOS (8050), papillary adenocarcinoma, NOS (8260), clear cell adenocarcinoma, NOS (8310), serous cystadenocarcinoma, NOS (8441), papillary serous cystadenocarcinoma, NOS (8460), serous surface papillary carcinoma (8461), Müllerian mixed tumour (8950), mesodermal mixed tumour (8951) and carcinosarcoma (8980) [24].

The trends in type I and II EC were analysed by joinpoint regression using the adjusted incidence rate and standard deviation [25]. The average annual percentage change (AAPC) and 95% confidence intervals were obtained using a logarithmic model. The following criteria were established: a minimum of 2 years at the end of the series until a change was detected and a minimum of 3 consecutive years between 2 change points. A maximum of 5 change points throughout the period was maintained as a criterion predefined by the programme.

Stata/SE (version 14.4) was used for the incidence analysis, and Joinpoint (version 4.6) was used for the analysis of incidence trends. Statistical significance was established at *p* < 0.05.

## 3. Results

The first recorded cases of type II EC in the Region of Murcia and in the Bas-Rhin area were reported in 1996 and 1997, respectively. In the 1998–2012 period, 3913 cases of endometrial carcinomas were recorded for the two populations. Of these, 3756 were classified as type I or II EC.

Within the 1998–2012 period, 3270 cases of type I endometrial carcinoma were recorded: 1764 (54%) in Murcia and 1506 (46%) in the Bas-Rhin area. In both populations, the largest number of cases were observed in the last study period (2008–2012).

A total of 486 cases of type II endometrial carcinoma were observed. Fifty-five percent occurred in the Bas-Rhin area, and the remaining 45% occurred in the Region of Murcia. Both in Murcia and in Bas-Rhin, the absolute number of cases increased in each five-year period.

Crude incidence rates of type I EC increased in both populations throughout the study period (Table 1). 

Women older than 65 years showed higher rates, while the group under 55 years had the lowest rates. In the Region of Murcia, women between 65 and 74 years accounted for the highest rates in all periods, and the opposite occurred in those under 55 years, with rates lower than six cases per 100,000 py. In the Bas-Rhin area, women older than 84 years accounted for the highest rates, and again, those younger than 55 years accounted for the lowest, with rates lower than one case per 100,000 py.

Similarly, for type II endometrial carcinoma, crude incidence rates increased in both populations for the entire period (Table 2).

The group 65 years of age and over accounted for the highest rates, and the group under 55 years old accounted for the lowest rates in all periods. In the Region of Murcia, women who were 65–74 years of age and 75–84 years of age accounted for the highest rates. In the Bas-Rhin area, women older than 84 years accounted for the highest rates of type II endometrial carcinoma, much higher than the rates for the other age groups, as occurred for type I endometrial carcinoma.

In the 1998–2012 period, the Bas-Rhin area presented age-adjusted incidence rates (2013 European Standard Population) higher than those in the Region of Murcia for both type I endometrial carcinoma (24.2 and 19.3 cases per 100,000 py, respectively) and for type II endometrial carcinoma (4.4 and 2.3 cases per 100,000 py, respectively) (Table 3, Figure 1). However, the difference between the two populations narrowed towards the end of the study period.

With respect to the trends in these tumours throughout the study period, we observed a slight growth trend in both populations for type I endometrial carcinoma, but the trend was not statistically significant. No change points were detected in the trends in this tumour (Table 4 and Figure 2). This increasing trend was slightly higher in the Region of Murcia (0.9% annual average) than in the Bas-Rhin area (0.3% annual average).

For type II endometrial carcinoma, there was an increase in the trends in both populations during the study period. The average annual growth of 11.8% in the Region of Murcia was statistically significant. In the Bas-Rhin area, an average annual increase of 3.9% was observed, but the increase was not statistically significant (Table 4 and Figure 3). The trend remained stable, with no change points throughout the period.

## 4. Discussion

In this study, one of the first to compare the incidence of type I and II endometrial carcinoma between two different European populations, an increase in the incidence rates for both types of endometrial carcinoma was observed in both regions. Although this increase was only statistically significant for type II EC in the Region of Murcia, the results obtained showed an increasing trend in EC in the fifteen years included in the study. The incidence rates for type I EC were much higher than the incidence rates for type II EC, a finding that is in agreement with other reports [6].

In the Region of Murcia, women aged 65–74 years accounted for the highest incidence of type I EC, while for type II EC, in the most recent period, women aged 75–84 years accounted for the highest incidence. In the Bas-Rhin area, women over 84 years of age accounted for the highest incidence rates during the entire study period, both for type I and type II EC. The results observed based on age might be due, in part, to the role of oestrogens in the natural history of the disease. In the postmenopausal stage, oestrogenic stimulation of the endometrial tissue occurs, which, without correct inhibition by progesterone, can produce uncontrolled tissue proliferation. In addition, in recent decades, the use of oestrogen-only and oestrogen–progestogen hormone replacement therapies has been standardised, especially since the first results of studies carried out within the Women’s Health Initiative were published in 2002 [26]. Oestrogen-only hormone therapy in peri- or post-menopausal women accounted for a 50% increase in the incidence of endometrial hyperplasia or carcinoma [6,27]. In addition, some studies, such as the one carried out by Boll et al. with Dutch women, highlight the influence of the introduction of oral contraceptives in the 1970s as a possible causal factor of the increase in the incidence of EC in this population [28]. These risk factors may explain the differences in incidence by age group in both populations. In Spain, the use of oral contraceptives began later because their sale was prohibited until use was legalised in 1978. This would explain why, in the data for recent decades, the incidence was higher among French older women [29,30]. Another possible explanatory factor could be that the under-diagnosis that occurs in older populations may be greater in Murcia than the Bas-Rhin area.

Type I EC exhibited a slightly increasing trend in the study period both in the Region of Murcia (0.9% APC) and in the Bas-Rhin area (0.3% APC). In other studies that analysed type I EC, such as that performed in Danish women by Evans et al., an increase in the annual trend was also observed [24]. For type II EC, we observed an increasing trend that was much higher than that of type I EC in both populations (Region of Murcia APC: 11.8%; Bas-Rhin APC: 3.9%). In a study by Tuxen et al., the APC in type II EC was 4.9% for the 1978–2014 period; however, for the 1996–2014 period, the slope of growth was much steeper, with an average annual growth of 6.4% [31]. These results should be interpreted with caution because of a lack of consensus in the literature on which histologies have to be considered within types I and II of EC. Therefore, this lack of uniformity in the inclusion criteria may hamper the comparison [24,32,33]. Regarding the changes that we found in the trends, we cannot completely rule out some effect of improvement in diagnostic and coding procedures. The change in lifestyles over the last decades, leading to greater exposure of women to EC risk factors, should also be considered [26,28]. Another exogenous factor that may have contributed to the increase in the trend is obesity, as it is one of the most important risk factors for endometrial cancer [34,35,36]. The mechanism described is an alteration in the endogenous metabolism of oestrogen, of which the effect is mediated by the increase in body mass index, especially in postmenopausal women. There is an increase in the rate of conversion of androgen precursors to oestradiol because of an increase in the activity of the aromatase enzyme in adipose tissue, leading to a hyper-oestrogenic state. This effect seems to be more intense in type I EC because it includes histological types, which largely depend on hormonal exposure. In Europe, the prevalence of obesity has tripled in recent decades [37,38]. The change in anthropometric characteristics of the population may influence the increase in the trend because this type of pathology is strongly influenced by the prevalence of obesity. Crosbie et al. published a meta-analysis with 24 cohort or nested case-control studies, which showed increased risk of EC linked to obesity [35]. In addition, in the same meta-analysis, the relative risk of developing endometrial carcinoma was studied by combining the two most important factors: obesity and the use of hormone replacement therapy. In women who are obese and who had not used oestrogen–progestogen therapy, the relative risk was much higher. Therefore, the use of combined hormonal therapy in women who have reached menopause in France in the 1970s and 1980s could explain the differences in the trend curves observed for women from both regions. The use of hormonal therapy with progestogens would have exerted a protective effect on French women. To interpret the increase in the trend in the Region of Murcia with respect to the trend in the Bas-Rhin area, it is necessary to take into account the bias of women who underwent hysterectomy. Several studies have taken into account the effect of hysterectomy when assessing the incidence of EC [31]. Those studies showed that the incidence of EC increased when rates were corrected by history of hysterectomy. As we mentioned, with the use of hormone replacement therapy, it is possible that, in European countries such as France, the practice of hysterectomy is more widespread than in Spain, resulting in a potential underestimation of the incidence rates in the Bas-Rhin area.

### Strengths and Limitations

The main strength of this study is that it is population-based; therefore, it includes virtually all endometrial cancer cases occurring in the study areas, minimising the selection bias observed in other studies. In addition, studying two populations from different countries introduces variability that gives greater consistency and external validity to the results. The procedures and classifications used follow international standards and allow for comparisons between two different populations. These indicators show the behaviour of the disease in the population and provide relevant information related to the risk factors of EC, which helps its clinical approach.

Among the limitations, we highlight the issues with respect to monitoring the long-term trends in type I and II EC because new histological codes have been added and used in recent years. In addition, not having a unanimous classification of these two types of tumours makes comparisons with other studies difficult. On the other hand, it is difficult to determine the progression or the prognosis of the disease based on the risk stratification without data on tumour stage. In future studies, it would be interesting to analyse variables that may influence the outcome, such as hysterectomy [39].

This study is expected to find differences in the behaviour of endometrial cancer in two populations from different areas of Europe, over a 15-year study period. Future studies could delve into the factors that may be the cause of this difference, in order to reduce the incidence of this cancer in the population. The impact in clinical practice would be to know the influence of the main risk factors that increase endometrial cancer.

## 5. Conclusions

The results of this study of EC incidence and trends in two populations indicate that incidence continues to increase in some populations, such as the Region of Murcia, whereas in others, there is a subtler increase and even slight stabilisation. The results support the importance of deepening the study of risk factors and their impact on different populations to understand this difference in rates and, if possible, to apply measures to slow and even reduce the risk of endometrial cancer. Type II endometrial carcinoma is a cancer that affects predominantly older women and, in addition to being more aggressive, has a worse prognosis. We have observed an increase in the annual trend for this type of carcinoma, and therefore, further studies to identify potential risk factors and efforts to improve recording and monitoring of EC cases are warranted.

## Figures and Tables

**Figure 1 ijerph-19-03789-f001:**
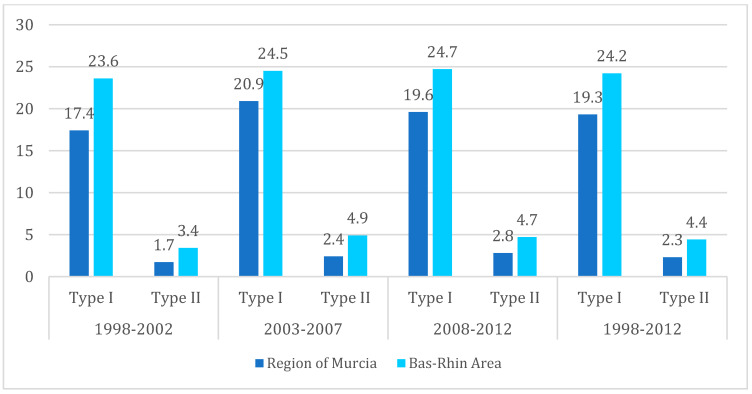
Age-adjusted incidence rates/100,000 person-years (2013 European Standard Population) for type I and II endometrial carcinoma, by period in the Region of Murcia and Bas-Rhin area, 19982012 period.

**Figure 2 ijerph-19-03789-f002:**
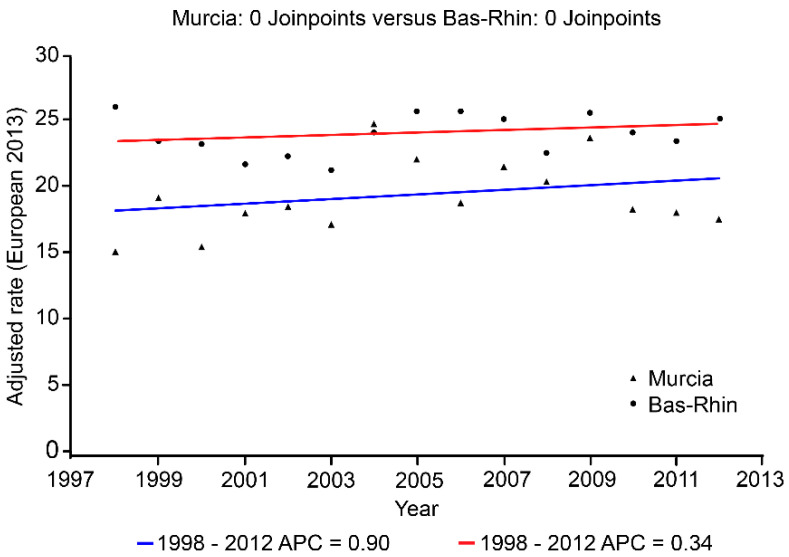
Joinpoint analysis of age-adjusted incidence rates (per 100,000) of type I endometrial carcinoma in the Region of Murcia and the Bas-Rhin area, 1998–2012 period. APC: Annual Percent Change.

**Figure 3 ijerph-19-03789-f003:**
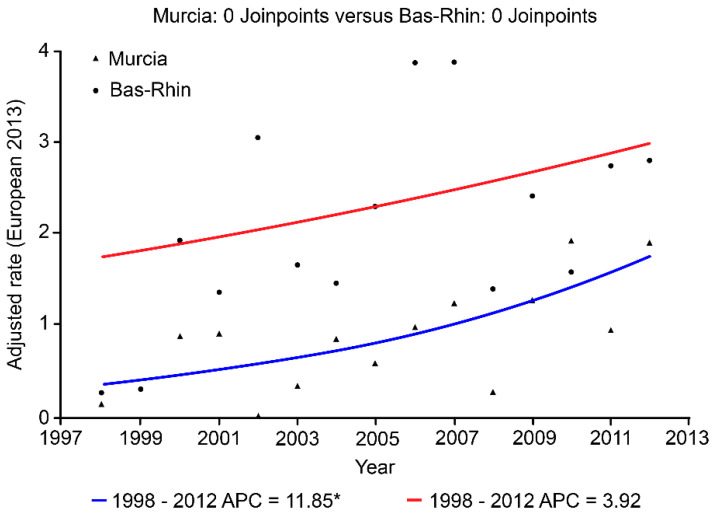
Joinpoint analysis of age-adjusted incidence rates (per 100,000 py) of type II endometrial carcinoma in the Region of Murcia and the Bas-Rhin area, 1998–2012 period. APC: Annual Percent Change. * *p* < 0.01.

**Table 1 ijerph-19-03789-t001:** Crude incidence rates/100,000 person-years for type I endometrial carcinoma, by age and period in the Region of Murcia (RM) and Bas-Rhin area, 1998–2012.

Period	1998–2002		2003–2007		2008–2012		1998–2012	
Age (Years)	RM	Bas-Rhin	RM	Bas-Rhin	RM	Bas-Rhin	RM	Bas-Rhin
<54	4.5	0.4	5.7	0.6	5.5	0.7	5.2	0.6
55–64	49.5	23.5	53	19.9	46.9	16.3	49.7	19.5
65–74	54.5	62	70	69.1	62.1	82.8	62.2	71.4
75–84	34	121.5	42.8	117.6	48	116	41.9	118.1
>84	7.2	240.7	12.3	267.5	12.1	239.3	10.9	248
<65	10.1	2.8	11.7	3	11.2	2.9	11	2.9
≥65	38.4	101.9	47.9	108.1	45.6	116.8	44.1	109.3
TOTAL	16	16.8	19	18.6	18.3	20.8	17.8	18.8

**Table 2 ijerph-19-03789-t002:** Crude incidence rates/100,000 person-years for type II endometrial carcinoma, by age and period in the Region of Murcia (RM) and Bas-Rhin area, 1998–2012.

Period	1998–2002		2003–2007		2008–2012		1998–2012	
Age	RM	Bas-Rhin	RM	Bas-Rhin	RM	Bas-Rhin	RM	Bas-Rhin
<54	0.1	0.0	0.4	0.0	0.2	0.1	0.2	0.0
55–64	5.2	1.7	3.9	2.8	4.8	0.6	4.6	1.6
65–74	4.3	4.8	9.3	10.5	12.2	11.3	8.8	8.9
75–84	6.6	19.8	9.8	23.4	9.7	18.9	8.8	20.7
>84	2.7	53.7	1.5	75.1	5.8	88.1	3.5	74.2
<65	0.8	0.2	0.8	0.3	0.8	0.1	0.8	0.2
≥65	4.9	15.3	7.9	22.1	9.8	24.8	7.7	21.0
TOTAL	1.6	2.3	2.3	3.6	2.7	4.0	2.2	3.3

**Table 3 ijerph-19-03789-t003:** Age-adjusted incidence rates/100,000 person-years (2013 European Standard Population) for type I and II endometrial carcinoma, by period in the Region of Murcia and Bas-Rhin area, 1998–2012 period.

Area	1998–2002		2003–2007		2008–2012		1998–2012	
	Type I	Type II	Type I	Type II	Type I	Type II	Type I	Type II
Region of Murcia	17.4	1.7	20.9	2.4	19.6	2.8	19.3	2.3
Bas-Rhin	23.6	3.4	24.5	4.9	24.7	4.7	24.2	4.4

Rate ratio between the Bas-Rhin (BR) and the Region of Murcia (RM).

**Table 4 ijerph-19-03789-t004:** Average annual percentage change (AAPC) in age-adjusted incidence rates (2013 European Standard Population) for type I and II endometrial carcinomas in women from the Region of Murcia and the Bas-Rhin area, 1983–2012 period.

Area	Period	Type I		Type II	
		AAPC (95%CI)	*p*-Value	AAPC (95%CI)	*p*-Value
Region of Murcia	1998–2012	0.9 (−1.01; 2.84)	0.33	11.8 (2.2; 22.4)	0.01
Bas-Rhin	1998–2012	0.34 (−0.48; 1.18)	0.39	3.9 (−2.8; 11.2)	0.23

## Data Availability

Data may be accessible upon reasonable request to the corresponding author.

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
