# Peer review of "Incidence and Trend of Type I and II Endometrial Cancer in Women from Two Population-Based European Cancer Registries (1998–2012)"

_ijerph, 2022, doi:10.3390/ijerph19073789_

Round 1

Reviewer 1 Report

I read with great interest the manuscript titled “Incidence of and Trends in Types I and II Endometrial Cancer, 2 in Women of two Different Populations in Europe, 1998-2012”, which falls whithin the aim of Int. J. Environ. Res. Public Health. The methodology is accurate, and the data analysis supports conclusions. However, the authors should clarify just a few points and revise the introduction.
Notably, the introduction does not include a brief surgical treatment for Endometrial Cancer. This point should be added after line 63, and the following three novel reviews should be cited:
- doi: 10.12688/f1000research.17408.1
- https://doi.org/10.1016/j.soncn.2019.02.002
- doi: 10.3389/fsurg.2021.790152

Secondly, the manuscript should be further revised by a native English speaker 

At least, I could not find any information regarding the approval of the Institutional Review Board. Did the authors have the approval before the study started? 

I suggested performing the major revisions for all these reasons.

Author Response

We thank the reviewers for their thorough evaluation of our manuscript and their insightful comments. We think the manuscript has improved significantly and hope the reviewers find this revised version satisfactory. Below we include a point-by-point response to all issues raised.

Response to Reviewer 1

Point 1: Notably, the introduction does not include a brief surgical treatment for Endometrial Cancer. This point should be added after line 63, and the following three novel reviews should be cited:

Response 1: We are deeply grateful to the reviewer for providing us with these references. We have included these data to introduce some features of endometrial cancer treatment (see Introduction section): “The majority of endometrial cancers receive surgical treatment only, currently by minimally invasive surgery (which sometimes includes the robotic technique). In a number of situations where the disease has expanded, adjuvant chemotherapy and radiotherapy are also indicated. However, survival has not improved in all cases. To this end, sentinel node lymph evaluation is one of the main options under study in the last decade, replacing generalized lymphadenectomy, as it offers the possibility of selecting patients who would benefit from chemotherapy treatment, assuming a less aggressive test [12-14].”

 Point 2: Secondly, the manuscript should be further revised by a native English speaker.

Response 2: The manuscript has been edited and revised by a professional English editing service (American Journal Experts). However, as the need to further revise the manuscript has been raised by several reviewers, we have again conducted a careful English style revision ir order to improve the quality of the writing. We hope the reviewer find this final version satisfactory.

 Point 3: At least, I could not find any information regarding the approval of the Institutional Review Board.

Response 3: Thanks for pointing this out. With regards to ethical issues, all data were collected as part of the regular activity of population-based cancer registries and fully anonymized prior to analysis. Informed consent is not required when working with anonymous data. We have now included an ethical statement paragraph to make clearer this point in the manuscript: “The study was conducted in accordance with the EU 2016/679 General Data Protection Regulation (GDPR) regarding the use of anonymized population data [23]. All data collected in the study database for incidence analysis were anonymous, and thus no further ethical approval was required”.

Reviewer 2 Report

I read with great interest the Manuscript titled “Incidence of and trends in types I and II endometrial cancer, in women of two different populations in Europe, 1998-2012”, which falls within the aim of IJERPH.    

In my honest opinion, the topic is interesting enough to attract the readers’ attention. Methodology is accurate and conclusions are supported by the data analysis. Nevertheless, authors should clarify some point and improve the discussion citing relevant and novel key articles about the topic.

  • Manuscript should be further revised by a native English speaker
  • -  The authors have not adequately highlighted the strengths and limitations of their study. I suggest better specifying these points - DISCUSSIONS:  What are the actual clinical implications of this study? it is important to report the results obtained by the authors in the context of clinical practice and to adequately highlight what contribution this study adds to the literature already existing on the topic and to future study perspectives  
  • - I could not find any information regarding the approval of the Institutional Review Board. Did author this approval before the study start?  
  • Recent data support the feasibility of sentinel lymph node identification in case of endometrial cancer, especially for early stage disease, and stress the necessity of an appropriate pre-operative mapping.

Author Response

We thank the reviewers for their thorough evaluation of our manuscript and their insightful comments. We think the manuscript has improved significantly and hope the reviewers find this revised version satisfactory. Below we include a point-by-point response to all issues raised.

Point 1: Manuscript should be further revised by a native English speaker.

Response 1: The manuscript has been edited and revised by a professional English editing service (American Journal Experts). However, as the need to further revise the manuscript has been raised by several reviewers, we have again conducted a careful English style revision ir order to improve the quality of the writing. We hope the reviewer find this final version satisfactory.

Point 2: The authors have not adequately highlighted the strengths and limitations of their study. I suggest better specifying these points.

Response 2: In response to your suggestion, we have now added some remarks in the strengths and limitations section.

Point 3: What are the actual clinical implications of this study? It is important to report the results obtained by the authors in the context of clinical practice and to adequately highlight what contribution this study adds to the literature already existing on the topic and to future study perspectives 

Response 3: We agree with the reviewer and we have now included the following paragraph in the Discussion section: “This study is expected to find differences in the behavior of endometrial cancer in two populations from different areas of Europe, over a 15-year study period. Future studies could delve into those factors that may be the cause of this difference, in order to reduce the incidence of this cancer in the population. The impact on clinical practice would be to know the influence of the main risk factors that increase endometrial cancer”.

Point 4: I could not find any information regarding the approval of the Institutional Review Board. Did author this approval before the study start?

Response 4: Thanks for pointing this out. With regards to ethical issues, all data were collected as part of the regular activity of population-based cancer registries and anonymized prior to analysis. Informed consent is not required when working with anonymous data. We have now included an ethical statement paragraph to make clearer this point in the manuscript: “The study was conducted in accordance with the EU 2016/679 General Data Protection Regulation (GDPR) regarding the use of anonymized population data [23]. All data collected in the study database for incidence analysis were anonymous, and thus no further ethical approval was required”.

Point 5: Recent data support the feasibility of sentinel lymph node identification in case of endometrial cancer, especially for early stage disease, and stress the necessity of an appropriate pre-operative mapping.

Response 5: We have added this new procedure in the introduction (see Introduction section): “To this end, sentinel node lymph evaluation is one of the main options under study in the last decade, replacing generalized lymphadenectomy, as it offers the possibility of selecting patients who would benefit from chemotherapy treatment, assuming a less aggressive test [12-14].”

Reviewer 3 Report

This study aims to analyze the long-term trends(15 years) in types I and II endometrial cancer in women in different European populations. This is a population-based study which included all women with endometrial carcinoma with minimal selection bias. As we know that molecular classification has now replaced former Bokhman binary classifications, it would be more interesting to analyze the incidence trend and risk factors of endometrial carcinoma based on molecular classification in future. There are several comments listed below:

1.I am curious why the time period between 1998 and 2012 was chosen for this study? Why the cases in recent years (after year 2012) were not included?

2.There is different incidence among different ages and there are many factors which might affect the incidence of endometrial carcinoma. I am wondering is it possible to do further analysis of the relations ship between risk-factors and the incidence of endometrial carcinoma? Is it possible to provide suggestions for women based on the results on the current study (such as oral contraceptives etc.)?

3.Line 303, "cervical cancer"? I assume it is “endometrial carcinoma”. Please check is seriously.

4.It might be better to update the references.

Thus, the data in this study is currently not sufficient to publish in this journal.

Author Response

We thank the reviewers for their thorough evaluation of our manuscript and their insightful comments. We think the manuscript has improved significantly and hope the reviewers find this revised version satisfactory. Below we include a point-by-point response to all issues raised.

Response to Reviewer 3

Point 1: As we know that molecular classification has now replaced former Bokhman binary classifications, it would be more interesting to analyze the incidence trend and risk factors of endometrial carcinoma based on molecular classification in future.

Response 1: We acknowledge the progress of molecular classification of endometrial cancer, as Charo ML et al. expose in their study: “Recent advances in endometrial cancer: a review of key clinical trials from 2015 to 2019” (this reference has now been included in the revised version of the manuscript), and agree that the molecular profile of a tumor can better define its prognosis and response to therapy than histology and stage alone. Nevertheless, the methods required for classification are currently quite expensive and require special tissue handling, which limits applicability. Routine molecular classification has not yet been adapted, and for this reason we used the Bokhman classification in the present study.

 Point 2: I am curious why the time period between 1998 and 2012 was chosen for this study? Why the cases in recent years (after year 2012) were not included?

Response 2: It has only been possible to include those cases that were already notified in both cancer registries, in order to be able to make a comparison between the two. The cancer registration process requires an expert evaluation; therefore, the data are collected with some inevitable delay.

Point 3: There is different incidence among different ages and there are many factors which might affect the incidence of endometrial carcinoma. I am wondering: is it possible to do further analysis of the relationship between risk-factors and the incidence of endometrial carcinoma? Is it possible to provide suggestions for women based on the results on the current study (such as oral contraceptives etc.)?

Response 3: It will be of great interest to analyze the relationship between risk factors and incidence. The present study allows to know that there are differences between age groups and between women from different populations that could reflect variability in the risk factors distribution. This work deals with possible explanations in the variability across regions taking into consideration differential risk factor exposure. The ecological design of the current study is useful to analyze correlations between incidence and risk factors. Further studies are needed with a prospective design to establish relationship between them. We have briefly speculated about potential implications for prevention of our results along the draft.

Point 4: Line 303, "cervical cancer"? I assume it is “endometrial carcinoma”. Please check is seriously.

Response 4: Thanks for the comment; we have corrected the error.

Point 5: It might be better to update the references.

Response 5: We have added three new recent articles in the Introduction section to update our study:

Charo, LM.;. Plaxe, SC.; Recent advances in endometrial cancer: a review of key clinical trials from 2015 to 2019. F1000Research 2019, 8, doi: 10.12688/f1000research.17408.1.

Cianci, S.; Arcieri, M.; Vizzielli, G.; Martinelli, C.; Granese, R.; La Verde, M.; Fagotti, A.; Fanfani, F.; Scambia, G.; Ercoli, A. Robotic Pelvic Exenteration for Gynecologic Malignancies, Anatomic Landmarks, and Surgical Steps: A Systematic Review. Frontiers in Surgery 2021, 8, 790152, doi: 10.3389/fsurg.2021.790152.

Passarello, K.; Kurian, S.; Villanueva, V. Endometrial Cancer: An Overview of Pathophysiology, Management, and Care. Seminars in oncology nursing 2019, 35(2), 157-165, doi: 10.1016/j.soncn.2019.02.002.

Round 2

Reviewer 3 Report

Thanks for your rrply to the questions.  We look forward further studies on the risk factors and EC in near futrue.